# Full-color processible afterglow organic small molecular glass

Yufeng Xue[1], Zongliang Xie [1,2], Zheng Yin[1], Yincai Xu[1] & Bin Liu [1,2] ✉

Organic afterglow materials, known for their unique luminescent properties and diverse applications, have garnered significant attention in recent years. However, developing long-lasting, high-efficiency, full-color afterglow systems and exploring simple materials processing strategies for new applications are still challenging in this field. Herein, we rationally design a processable molecular glass and employ it as a host in a host-guest strategy to address these challenges. By strategically modifying the host via othyl-methylation, we successfully create a molecular glass and capture its temperature-dependent, processable viscous supercooled liquid state. High-efficiency full color from violet to near-infrared afterglow systems with ultralong lifetimes are developed by doping varied structural dopants. The underlying glass-forming and afterglow mechanisms are also clearly elucidated and verified. Moreover, the excellent glass-forming ability of the host and its viscous supercooled liquid enabled the glass system for large-area fabrication, shaping of objects with diverse 3D structures, and creation of flexible, meter-long afterglow fibers. This work offers significant potential for practical applications in advanced textiles, displays, and other fields.

Organic afterglow materials, capable of maintaining luminescence for over several tens of milliseconds or even seconds after the excitation ceases, have garnered significant attention in recent years for potential applications in fields such as multicolor display, encryption, bioimaging, and responsive sensors[1–5]. To date, organic afterglow has been achieved by various strategies such as host-guest doping strategy[6–10], crystal-inducement[11–14], supramolecular assembly[15,16], co-crystallization[17–19], H-aggregation[20,21] and others. However, developing materials that exhibit long-lasting, full-color afterglow from violet to near-infrared while achieving high phosphorescence quantum yields remains challenging. Additionally, finding simple processing methods for new applications is still a hurdle in this field. While small molecules offer advantages like precise synthesis, easy purification, and batch-to-batch homogeneity, their tendency to crystallize and the weak intermolecular interactions hinder large-area fabrication, long fiber production, and complex 3D structure formation. To enhance the processability of small molecules in practical applications, they have to be fabricated into uniform particles and then mixed with matrices such as aloe vera gel[22,23], UV curable

resin[24,25]. However, this approach increases the molecular surface area, making them more susceptible to environmental quenching from oxygen and moisture. On the other hand, disrupting the crystal lattice can reduce vibration suppression, potentially weakening the afterglow performance[26]. Given these limitations, it is crucial to establish a versatile and easily accessible system with tunable afterglow properties and excellent processability to fully unlock the application potential of organic afterglow materials.

Host-guest systems are more likely to address both challenges simultaneously, as guests play a crucial role in fine-tuning emission color and brightness, and the host influences microenvironment rigidity, mechanical properties[27], and processability[28]. For the host design, high triplet energy level is also expected to avoid energy back transfer from the guest after excitation. However, phase separation during doping, especially when dopant and host structures differ significantly, can lead to non-uniform or even the failure to produce afterglow. Molecular glasses (MGs) are a type of organic small molecules that can form an amorphous solid at room temperature and a

[1]Department of Chemical and Biomolecular Engineering, National University of Singapore, Singapore, Singapore. [2]Institute for Functional Intelligent Materials, National University of Singapore, Singapore, Singapore. ✉e-mail: cheliub@nus.edu.sg

viscous supercooled liquid between the glass transition temperature and the melting point, offering a promising solution for improving the dopant tolerance and processability of small molecular afterglow system[29–33] (Fig. 1a). From the structure perspective, MGs typically have irregular structures leading to multiple conformations which help hinder the rearrangement of molecules, thereby reducing the crystal growth rate, and enabling the existence of a temperature-dependent, processible viscous supercooled liquid state[33]. Based on the above analysis, if an irregularly shaped molecule provides a rigid environment, it could serve as a processible glassy afterglow host.

Triphenylphosphine oxide (TPPO) has been used as an afterglow host due to its rigid tetrahedron structure and high triplet energy level[10,34–36]. However, severe phase separation occurs in systems when the host and guest have significant structural differences. By introducing three methyl groups to disrupt the symmetry of the phenyl group, we developed a new host molecule, tri(2-methylphenyl) phosphine oxide (TTPO), which retains the strong afterglow performance of TPPO while significantly slowing down the crystallization rate and exhibiting temperature-dependent viscous supercooled liquid behavior (Fig. 1b). TTPO MG could be easily prepared on a hundred-gram scale (Fig. 2a). Dopants at 1 wt% with various structures were uniformly dispersed, producing full-color afterglow with maximum emissions ranging from 410 to 767 nm and lifetimes varying from 3 to 1695 ms. The glass-forming ability of TTPO allows for large-area preparation and the presence of viscous supercool liquid endows us opportunities to process the glass system in a manner like inorganic glass. Objects with diverse 3D structures can be easily shaped by processing the viscous supercooled liquid. Additionally, for the first time, flexible meter-long fibers with afterglow waveguide properties have been successfully fabricated from the small MGs. This work presents a valuable strategy for developing organic afterglow materials with excellent processability, offering significant potential for practical applications in wearable devices, smart displays, flexible electronics, and other fields.

## Results

To demonstrate our molecular glass design strategy, TTPO was firstly prepared from tris (2-methylphenyl) phosphine (TTP) on a hundred-gram scale by using $H_2O_2$ as an oxidizer. The low-cost reactants, mild reaction conditions (1 h, 99% yield), and simple purification via recrystallization make this reaction atom-economic to yield cost-effective TTPO (Fig. 2a). The methyl group introduced at the *ortho*-position of three phenyl rings resulted in the symmetry disruption of the phenyl group which led to conformational diversity. Four distinct energy conformations have been predicted[37,38], with *exo₃* being the most stable conformation, as indicated by the single crystal structure.

The least stable *exo_O* conformation exhibits a 33.5 kJ·mol⁻¹ higher relative energy compared to that of *exo₃* (Fig. 2b). To arrange disordered molecules into an ordered and stable conformation, interconversion barriers must be overcome, which slows down the crystallization rate and results in a glass transition temperature at 299 K (Supplementary Fig. 1). On the other hand, the tetrahedral structure led to weak conjugation between the phosphine oxide and phenyl groups. As a result, methylation does not alter its triplet energy levels compared to TPPO (Supplementary Fig. 2), which ensures that TTPO is suitable as a good phosphorescent host in terms of energy levels ($T_1 = 2.95$ eV).

Glass forming is a kinetic process where solids form before molecules are arranged into order. Therefore, vitrification is the key step in creating a disordered microstructure. By rapidly cooling samples from the melted state (433 K) to room temperature, a uniform and transparent film was easily produced. In contrast, allowing the sample to cool naturally resulted in the formation of a crystalline structure (Supplementary Fig. 3). To assess the glass quality, Kissinger analysis[39,40] was used to estimate the overall apparent activation energy of crystallization from amorphous glass (Eq. (1)).

$$\ln\left(\frac{\beta}{T_p^2}\right) = \ln\left(\frac{AR}{E_a}\right) - \frac{E_a}{RT_p} \tag{1}$$

In Eq. 1, $\beta$ is the heating rate in K·min⁻¹, $T_p$ is the peak crystallization temperature where the crystallization rate reaches the maximum, $A$ is the frequency factor in s⁻¹, $R$ is the universal constant (8.314 J·mol⁻¹·K⁻¹), and $E_a$ is the overall apparent activation energy from glass to crystal in J·mol⁻¹. By plotting the natural logarithm of heating rates against the maximum rate of crystallization temperature ($T_p^{-1}$), a linear relationship can be observed, where the overall activation energy was calculated as $47 \pm 2$ kJ·mol⁻¹, resulting in moderate morphology stability (Fig. 2c, Supplementary Fig. 5). The Kissinger analysis demonstrates the effectiveness of *o*-methylation in enhancing TTPO's glass-forming ability.

A key characteristic of MG is the presence of a temperature-dependent viscous supercooled liquid state. Rheological measurements were conducted using a rotational rheometer, clearly demonstrating that the melting liquid and supercooled liquid exhibit temperature-dependent viscosity behavior (Fig. 2d). A shear deformation at a rate of 30 s⁻¹ was applied to the melting liquid viscosity at 443 K and the supercooled liquid at 358 K. The melting liquid viscosity is 0.03 Pa·s, while the fragile supercooled liquid undergoes shear-induced crystallization, a process triggered by heterogeneous nucleation at the interface between the upper plate and the sample

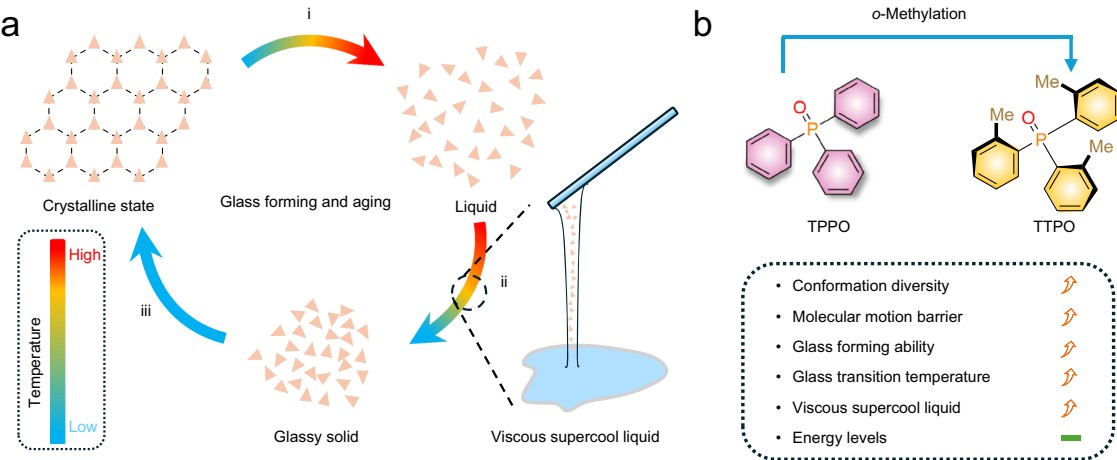

**Fig. 1 | Glass forming cycle and glass design strategy. a** Illustration of glass forming and aging. (i), Melting (ii), vitrification (iii), molecule reorganization. **b** *O*-methylation from TPPO to TTPO.

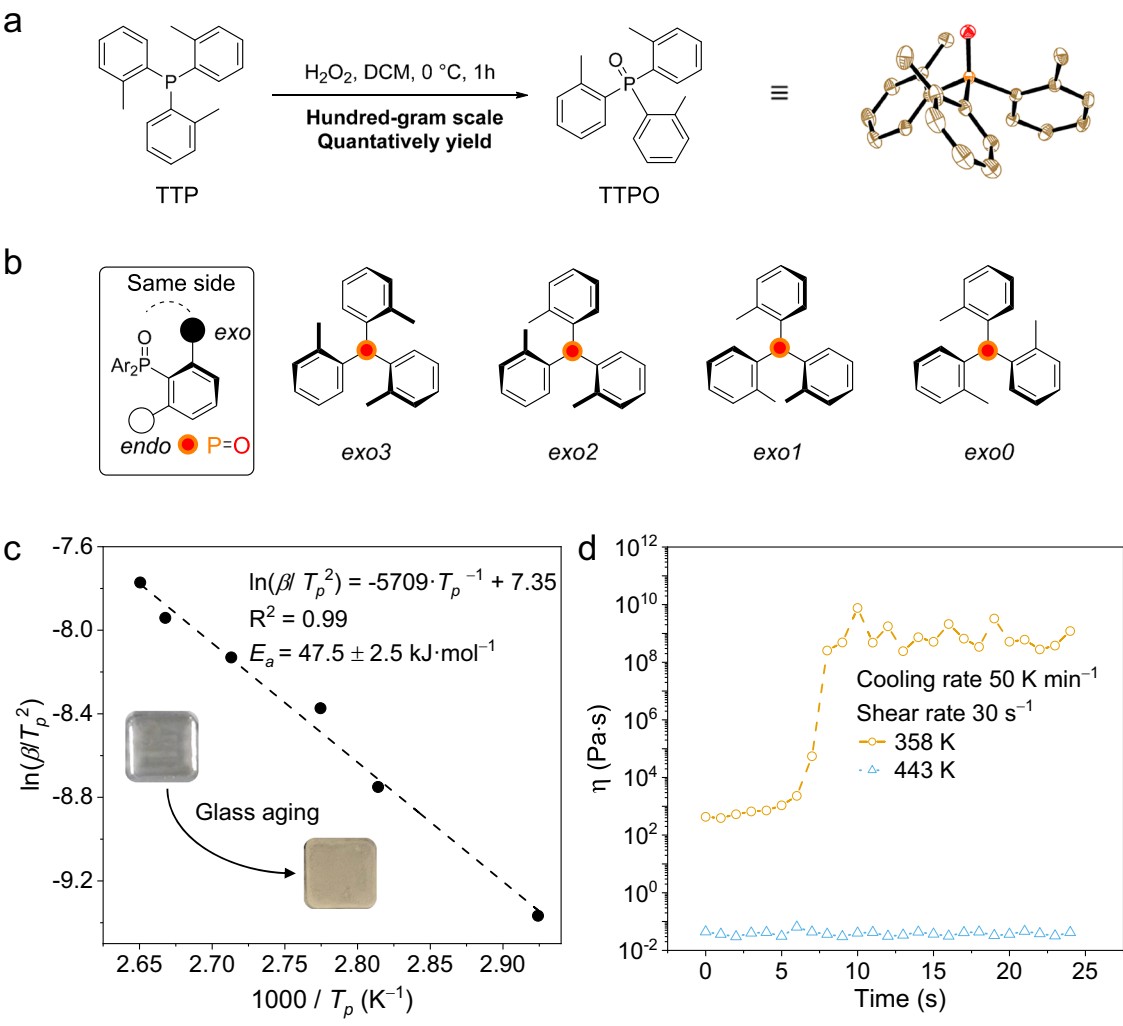

**Fig. 2 | Glass properties of TTPO. a** Synthetic route to TTPO. **b** TTPO conformations. **c** Kissinger analysis to determine morphology stability. **d** Viscosity of melt liquid at 443 K and supercool liquid at 358 K.

during viscosity measurement. From the initial stage to 6 s, the viscosity of supercool liquid increased from 379 Pa·s to 1057 Pa·s. It's important to note that the viscosity remains within the working range and is suitable for processing[30,41,42]. After 7 s, rapid crystallization was observed, indicated by a significant increase in viscosity from 1989 Pa·s to $10^9$ Pa·s. Although the supercooled liquid tends to crystallize under disturbances, its viscous nature allows us to process it similarly to inorganic glass.

From the crystal analysis, TTPO is tightly packed due to a variety of intermolecular interactions, including numerous C−H···O (2.6-2.9 Å), C-H···π (2.9 Å) and C-H···H (2.4 Å) interactions (Fig. 3a). Additionally, the presence of methyl groups introduces motion barriers, further restricting molecular movement and contributing to a rigid environment within the crystal. Considering its high triplet energy level, TTPO has the potential to be a room-temperature phosphorescence (RTP) host. As proof of the concept, twist molecule tris(1-naphthyl) phosphine (TNpP) was doped into TTPO at a 1 wt% concentration by slowly evaporating the solvent. The resulting crystalline solid exhibited a bright yellow-green afterglow with a lifetime of 270 ms after 3 s of photoactivation (365 nm, 5 W, UV lamp) as shown in the Fig. 3d. In contrast, no RTP was observed when the same excitation was applied to pure TNpP. The RTP emission from the doping system is in line with the phosphorescence of TNpP@Toluene at 77 K, instead of TTPO powder, therefore, we confirmed that the RTP emission originated from TNpP rather than TTPO (Supplementary Fig. 16). To test whether

glassy TTPO could maintain the same stabilization effect as powder form, glassy samples with the same doping concentration were prepared. Following the rapid cooling method, we managed to create uniform and transparent rigid glassy solid systems without experiencing phase separation. The rigid glassy solid systems exhibit up to 87% transmission in the visible light range while absorbing light in the UV range, depending on the absorption characteristics of each dopant and the TTPO host itself (Fig. 3b). We found a photoactivation behavior when we test the RTP performance of glassy films. Phosphorescence remains inactive without photoactivation due to the inevitable presence of residual oxygen in the system during the fabrication of glass films in the air. As a result, upon continuous irradiation for 10 s, the triplet emission component in the TNpP@TTPO doping system significantly increases. This process was captured through time-dependent PL measurements (Supplementary Fig. 7a, b). After photoactivation, clear and bright afterglow could be observed from the TNpP@TTPO system in its amorphous form, as confirmed by *p*-XRD experiments (Fig. 3c). This phenomenon can be attributed to the generated triplet excitons from the guest molecules gradually consume the residual ground-state oxygen, thereby activating the phosphorescence (Supplementary Fig. 7c). Remarkably, the resulting transparent and uniform TTPO doping glass exhibited nearly identical photophysical properties (Fig. 3d and Supplementary Fig. 8). This consistency indicates that the rigid environment is effectively maintained in the amorphous state. The environmental stability of the

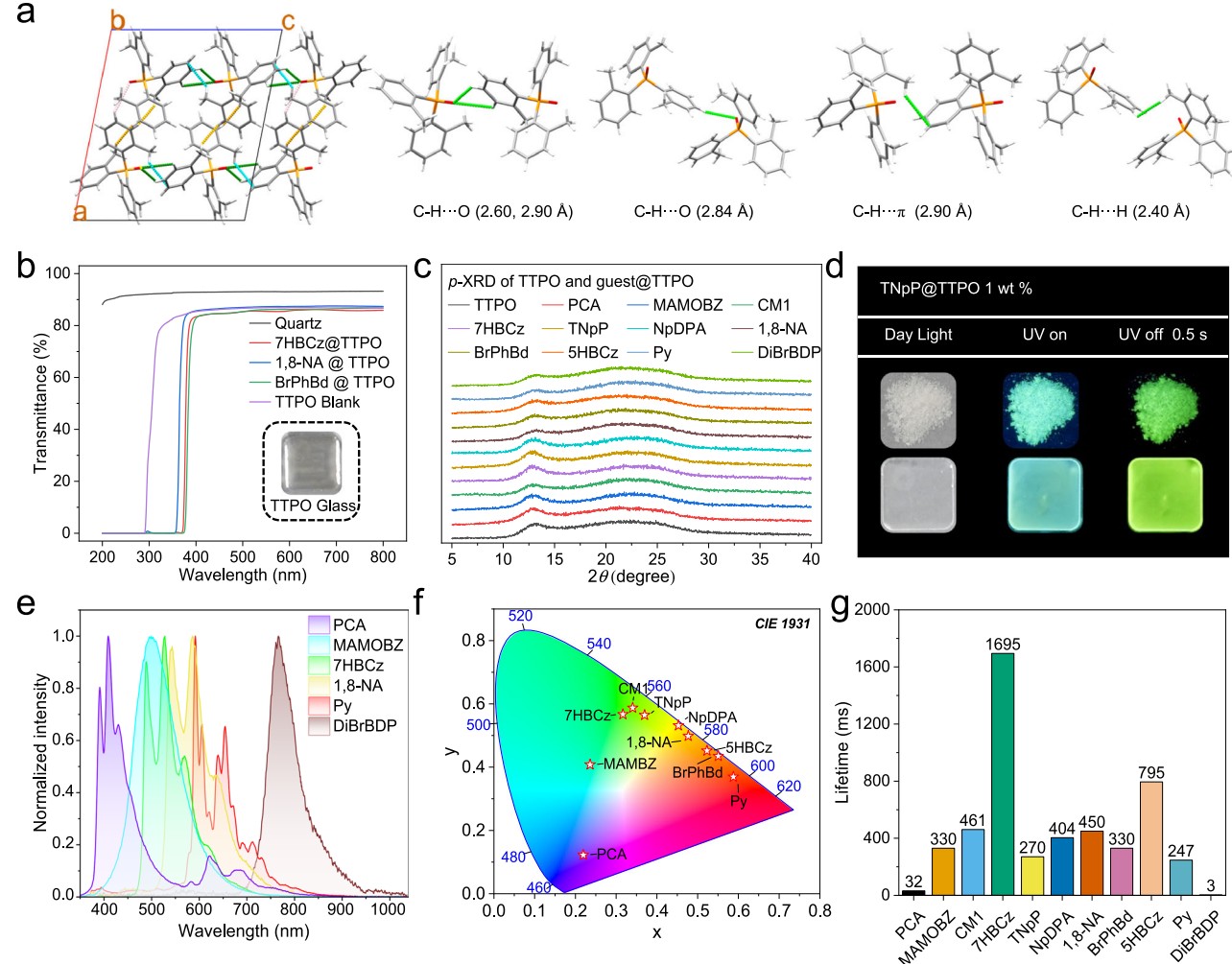

**Fig. 3 | Intermolecular interactions in TTPO crystal and afterglow glass properties of different doping systems. a** Intermolecular interactions in TTPO crystal. **b** Transmission of the quartz substrate, blank TTPO glass, and different doping systems. **c** p-XRD of doping systems after excitation. **d** TNpP@TTPO in powder form and glassy form. **e** Dealy 8 ms spectra of different TPPO doping systems. **f** Commission Internationale de L'Eclairage (CIE) 1931 diagram of different TTPO doping systems. **g** Lifetime of different doping systems. All spectra were collected under 365 nm excitation. Carema setting: ISO 800 (UV on), ISO 5000 (UV off).

TTPO doping systems was evaluated by immersing TNpP@TTPO films in pure water and exposing them to a pure oxygen atmosphere, respectively. After photoactivation, the TNpP@TTPO films exhibited bright phosphorescence with a duration of ~1.5 s in the water or oxygen. Notably, even after being submerged in water for three days, the afterglow performance remained largely unaffected (Supplementary Fig. 9). This further demonstrates that the TTPO film can effectively block quenching factors such as oxygen and humidity. The long-term stability of phosphorescence was also assessed, revealing that the phosphorescence intensity of the TNpP@TTPO film remained unchanged over nine weeks (Supplementary Fig. 10a, b). Furthermore, we evaluated the phosphorescence behavior in different morphological states by comparing the delayed spectra and the lifetime at 528 nm for both crystalline and glassy films. The results demonstrated that the phosphorescence properties remained consistent regardless of the morphological state (Supplementary Fig. 10c, d).

To demonstrate the excellent dopant tolerance of TTPO, a variety of molecules were selected as dopants and blended with TTPO at 1 wt%. This doping ratio was chosen because the TNpP@TTPO system exhibited the best performance at 1 wt% (Supplementary Figs. 11, 12). These included simple and small size molecule methyl 2-amino-3-methoxybenzoate (MAMOBZ), organic ion dye 2,6-dibromo-4,4-difluoro-1,3,5,7-tetramethyl-8-phenyl-4-bora-3a,4a-diaza-s-indacene

(DiBrBDP), molecules with planar structures, such as coumarin 1 (CM1), 7H-benzo[c]carbazole (7HBCz), 5H-benzo[b]carbazole (5HBCz), 1,8-naphthalic anhydride (1,8-NA) and pyrene (Py), as well as twisted structures like (diphenyl)(1-naphthyl)amine (NpDPA) (Supplementary Fig. 13). After employing the melting-rapid cooling method, the doping systems could form a uniform and transparent glassy state without phase separation, as confirmed by p-XRD (Fig. 3c). Following 10 s of photoactivation, all doping systems exhibited highly efficient afterglow, ranging from cyan to near-infrared with lifetimes varying from 3 ms to 1695 ms (Figs. 3e–g, 4a, Supplementary Fig. 20). These delayed emissions differed from the fluorescence of the guest molecules but were consistent with their phosphorescence at 77 K (Supplementary Fig. 16). This indicates that the doping system exhibited RTP arising from the triplet-state emission of the guests themselves. Notably, DiBrBDP demonstrated a near-infrared afterglow with a maximum emission at 767 nm with a 3 ms lifetime. In the Py@TTPO system, red emission was observed with a maximum of 592 nm and a lifetime of 247 ms. An invisible strong delayed fluorescence at 397 nm was detected in the Py@TTPO system, which was confirmed to originate from the triplet-triplet annihilation (TTA) process of pyrene (Supplementary Fig. 21). However, most aromatic compounds have low triplet energy levels due to the extensive delocalization of π-electrons across their conjugated systems, and significant overlap between the highest

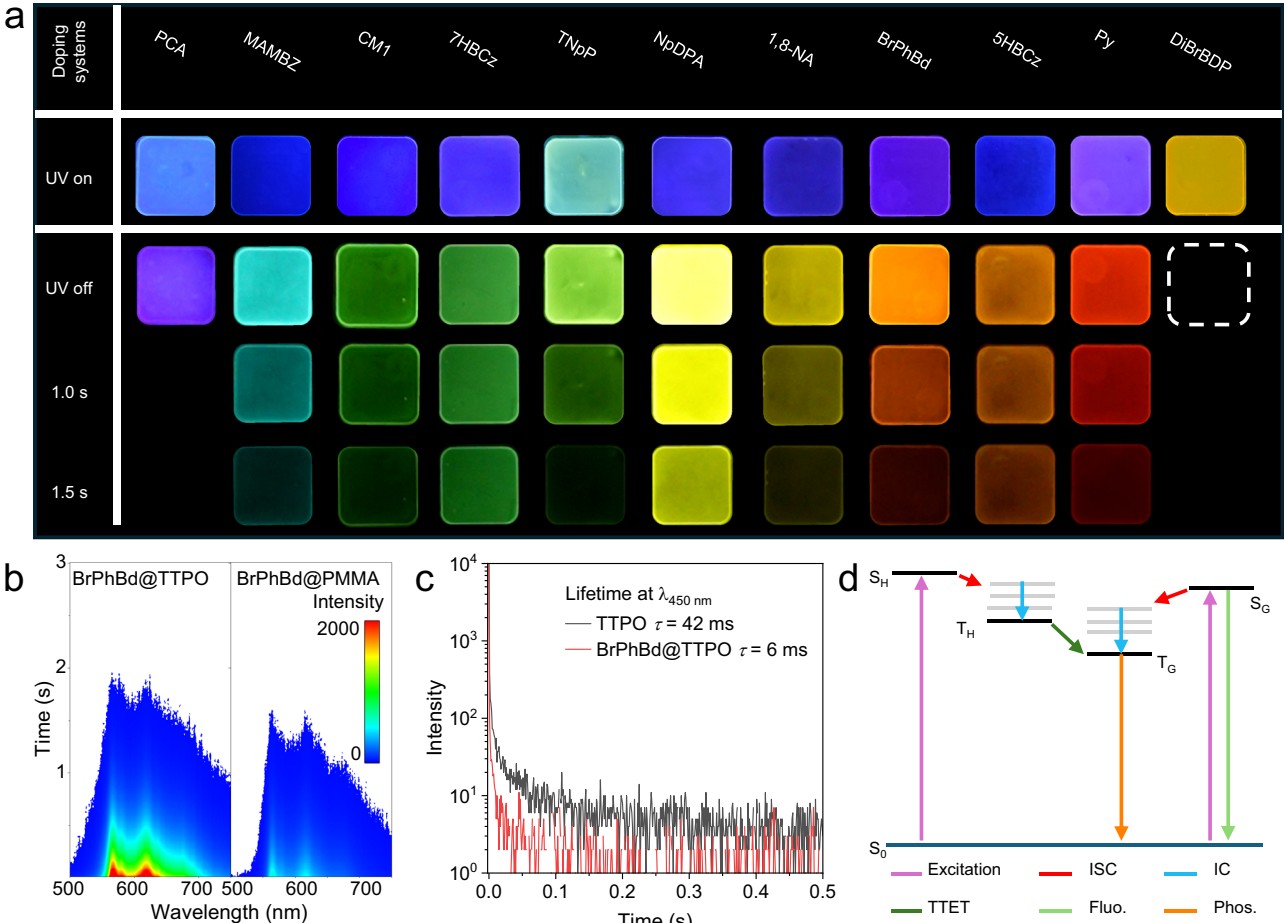

**Fig. 4 | Mechanism study of different doping systems. a** Photo of different doping systems. **b** TRES comparison of afterglow properties between BrPhBd@TTPO and BrPhBd@PMMA. **c** The lifetime at 450 nm of TTPO film and BrPhBd@TTPO doping system. **d** Jablonski diagram for proposed photophysical processes of doping systems. ISC intersystem crossing, TTET triplet-triplet energy transfer, IC internal conversion, Fluo. fluorescence, Phos. phosphorescence.

occupied molecular orbital (HOMO) and lowest unoccupied molecular orbital (LUMO). This overlap results in large singlet-triplet energy gaps ($\Delta E_{ST}$), as the $\Delta E_{ST}$ is proportional to twice the electron exchange energy which is determined by the degree of HOMO-LUMO orbital overlap[43]. To date, strategies for achieving deep blue RTP are still rare[44,45]. Herein a simple violet afterglow is realized by embedding 1-pyrene carboxylic acid (PCA), a dopant with greater conjugation than pyrene, into the amorphous TTPO matrix. This design leverages the up-conversion emission characteristics of the TTA process, producing a violet afterglow with a strong delayed fluorescence at 410 nm and a weak phosphorescence at 623 nm. The photophysical properties of guest@TTPO are summarized in Supplementary Table 1, indicating that afterglow systems exhibit a low non-radiative decay rate ($k_{nr}$) from 0.58 to 3.95 s$^{-1}$, supporting its role as an excellent afterglow host for diverse structural dopants.

To further investigate the luminescence mechanism, we doped the guest molecules into PMMA using annealing methods, which facilitate the formation of small aggregates that significantly enhance RTP and prolong lifetimes compared to unannealed systems[46]. The guest@PMMA systems exhibit weaker afterglow compared to the guest@TTPO systems under the same excitation conditions (Fig. 4b, Supplementary Fig. 23). A rigid environment typically restricts molecular vibration and reduces thermal disturbances in excited-state molecules, which may induce fine structure of the spectrum[47]. By comparing the afterglow spectra of guest molecules in both systems, clearer fine spectral structures were observed in guest@TTPO,

matching the phosphorescence spectra in toluene at 77 K. This confirms that TTPO provides a highly rigid environment, effectively suppressing non-radiative decay processes.

Beyond its rigid nature, TTPO also acts as an energy transfer platform to enhance afterglow[10,48–50] with its lowest triplet energy level ($T_1$ = 2.95 eV) positioned between the singlet ($S_1$ 2.88 to 3.56 eV) and triplet states ($T_1$ 2.08–2.86 eV) of the guest molecules (Supplementary Fig. 18). Through careful investigation the photophysical properties of the TTPO film, we found that the TTPO film exhibited a weak absorption at 365 nm, indicating that it is excitable under the excitation conditions of our doping system (Supplementary Fig. 24a). Photophysical studies revealed that the TTPO film exhibits fluorescence at 400 nm and weak but detectable phosphorescence at room temperature after photoactivation, with a maximum emission at 530 nm (Supplementary Fig. 24b). To investigate the triplet-triplet energy transfer (TTET) process, we measured the lifetime at 450 nm of the TTPO film (42 ms). Then compared this with the emission lifetime at 450 nm across various doping systems, where the guest phosphorescence spectrum does not exhibit emission at this wavelength. Taking BrPhBd@TTPO as an example, the lifetime of BrPhBd@TTPO at 450 nm is 6 ms, which is significantly shorter compared to TTPO itself (Fig. 4c). Additionally, microsecond TRES experiment for BrPhBd@TTPO was conducted. From Supplementary Fig. 25, the emission peak at 0.2 ms is attributed to the triplet emission of TTPO as its emission profile is identical to that of TTPO film. Between 0.2 ms and 0.4 ms, the emission undergoes a gradual redshift and

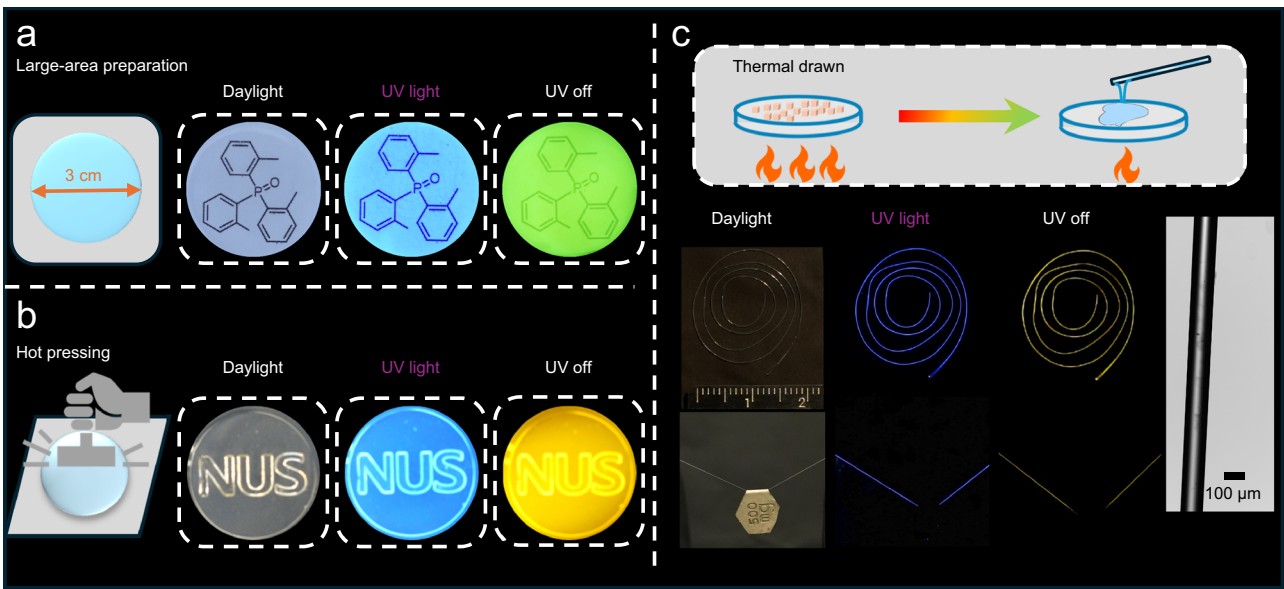

**Fig. 5 | Processing demonstration. a** Large-area preparation. **b** Hot pressing. **c** Thermal drawn. All materials were excited by a 365 nm UV lamp.

broadening. From 0.4 ms to 50 ms, the emission intensity from the TTPO triplet state progressively diminishes, while the triplet emission from BrPhBd gradually increases. This dynamic evolution provides strong evidence supporting the TTET process. Similarly, other doping systems, including 1,8NA@TTPO, NpDPA@TTPO, Py@TTPO, and DiBrBDP@TTPO, exhibit the same trend of reduced lifetime at 450 nm compared to the TTPO fim, also confirming the occurrence of TTET (Supplementary Fig. 26). Further theoretical calculations based on the guest molecules and their corresponding guest@TTPO pairs were carried out[49,51,52], and the results are shown in Supplementary Figs. 27–41. As exemplified by BrPhBd@TTPO, additional intersystem crossing (ISC) channels arose from host-guest pairing but were not present in the single BrPhBd molecule. As a result, multiple ISC channels, such as $S_1 \rightarrow T_5$ ($\Delta E_{ST} = 0.13$ eV; SOC = 0.02 cm$^{-1}$), $S_1 \rightarrow T_6$ ($\Delta E_{ST} = 0.15$ eV; SOC = 0.06 cm$^{-1}$), $S_1 \rightarrow T_7$ ($\Delta E_{ST} = 0.16$ eV; SOC = 0.22 cm$^{-1}$) are activated in the region of $\Delta E_{ST} \leq 0.3$ eV (Supplementary Fig. 37). Analysis from the newly generated triplet levels, electron–hole densities of $T_6$ and $T_7$ were localized on the TTPO component, although this process was less efficient compared to the transition from the guest's $S_1$ to $T_n$ state (Supplementary Fig. 38). However, no additional ISC channels were activated in the DiBrBDP-TTPO pair from the calculation results due to the mismatched energy between TTPO and DiBrBDP (Supplementary Fig. 43). In summary, a triplet-triplet energy transfer process is involved in the luminescence mechanism to boost the formation of triplet excitons and improve the RTP performance for these guest@TTPO systems (Fig. 4d). This enhancement highlights the pivotal role TTPO plays not only in providing a rigid environment but also in actively supporting energy transfer processes that lead to superior afterglow properties.

Upon vitrifying the TTPO melting liquid to room temperature, the supercooled liquid gradually becomes more and more viscous until it solidifies completely. The presence of a viscous supercooled liquid allows us to follow a processing method like that used for inorganic glass. For example, large-area preparation is desirable for display applications, as it ensures consistent performance across larger surfaces—an essential factor for commercial viability and the widespread adoption of consumer products. Through the vitrification technique, we were able to create a uniform and transparent afterglow glass with a 3.0 cm diameter (Fig. 5a). This high transparency and large area preparation ability make the afterglow glass promising for optical device development. The viscous state also enables us to shape materials into

desired forms. By hot-pressing the viscous supercool liquid, we successfully created an NUS stamp with a 3D structure (Fig. 5b, Supplementary Fig. 44). Beyond hot-pressing, the long-lasting viscous supercooled liquid offers the potential for thermal fiber drawing. We managed to produce a meter-scale fiber (0.5 m long with 100 ± 10 μm in diameter (Figs. 5c), 1 m long (Supplementary Fig. 45) simply through thermal drawing from the viscous supercool liquid. To the best of our knowledge, this is the longest afterglow fiber made purely from small organic molecules, as only a limited number of fibers based on organic small molecules can reach centimeter-level[53], with most still at the micron or even nano level. Due to the isotropic and uniform nature of glass, weak and dispersive interactions are distributed evenly in three orthogonal directions, therefore fibers demonstrate remarkable flexibility and mechanical strength (Supplementary Fig. 46). The coiled fiber can be stretched into a straight configuration, and a 0.5 mg fiber can support weights of up to 500 mg without breaking. The successful fabrication of long and flexible fibers opens up opportunities to unlock the potential applications of TTPO. To further explore its capabilities, we investigated its feasibility for optical waveguide fiber fabrication. A 7 cm long fiber was thermal drawn from the viscous supercool liquid of TNpP@TTPO system. Upon UV excitation at either the tip or the middle of the fiber, the fiber emits light at the incident end, with the emission propagating along the fiber axis, demonstrating waveguiding behavior. After the excitation source is removed, the fiber continues to emit light at its ends, which can be easily observed with the naked eye (Supplementary Fig. 47). Bright afterglow optical fibers are rarely reported, and in this system, TTPO serves as a transparent fiber host while the guest molecules provide luminescence properties. By modifying the phosphorescent dopants, the afterglow fiber characteristics can be further tuned and customized, offering a more flexible approach to exploring the potential applications of afterglow optical fibers in areas such as optical signal transmission, flexible photonic devices, and nighttime displays.

In conclusion, we have developed an effective strategy for designing an afterglow molecular glass host. The designed TTPO MG can be readily produced on a hundred-gram scale. The introduction of rigid methyl groups disrupts the symmetry of the phenyl group, leading to diverse conformations, excellent glass-forming ability, and enabling a temperature-dependent, processable viscous supercooled liquid state. The efficient afterglow is achieved using a host-guest strategy, producing full-color emission ranging from violet to near-

infrared with an ultralong lifetime. TTPO afterglow systems can be fabricated into a uniform, transparent film within one minute, from melting to cooling. The low-viscosity melt and small molecular size of TTPO facilitate the dissolution of most organic molecules. Additionally, the amorphous nature of TTPO at room temperature prevents phase separation in systems with significant host-guest structural differences. Based on the investigation of the luminescence mechanism, TTPO not only offers a highly rigid microenvironment but also functions as an energy transfer bridge, facilitating the generation of afterglow. As a result, TTPO demonstrates strong potential as a versatile afterglow host. These advantages endow TTPO with great potential to act as an excellent platform for the rapid screening of phosphorescent molecules. Additionally, the existence of a supercooled liquid state allows the glass system to be processed similarly to inorganic glass. Objects with 3D structures and meter-long fibers are fabricated without losing their afterglow properties. With its advantages in cost-effective large-scale synthesis, tolerance to various dopants, and high processability, TTPO-based afterglow systems hold great potential for new applications and the design of innovative devices.

## Methods

### General methods

Proton, carbon and phosphorus nuclear magnetic resonance ($^1$H NMR, $^{13}$C NMR, $^{31}$P NMR) spectra for the TTPO were attained from a Bruker ARX 400 NMR spectrometer with Chloroform-$d$ (CDCl$_3$). The Kissinger analysis for TTPO activation energy from glassy to crystalline state was determined by using a differential scanning calorimeter (PerkinElmer 8000 DSC). Thermogravimetric analysis of TTPO was performed by using a thermogravimetric analyzer (Shimadzu DTG-60 AH). The viscosity of TTPO supercool liquid was measured by using a rheometer (Anton Paar MCR-9). UV–vis absorption spectrum of each compound was obtained using a UV-vis spectrometer (Shimadzu UV-2600). Delayed emission spectra were collected with the Ocean Optic QE 65 Pro spectrometer. Photoluminescence (PL) spectra and phosphorescence lifetimes were recorded with an Edinburgh Instruments Spectrofluorometer (FLS 1000). The quantum yields of the doped systems were measured by FLS 1000 Spectrofluorometer with an integrating sphere. The crystal structure of TTPO is available from the CCDC database (CCDC number: 2425812).

### Theoretical calculations

Density functional theory (DFT) computations, encompassing geometrical optimization and electronic properties evaluations at ground states, were conducted using the Gaussian 09 software package at the B3LYP/6-311 g(d) level. The energy levels of the excited states were computed using the time-dependent DFT (TDDFT) method at the same B3LYP/6-311 g(d) level. The spin-orbit coupling (SOC) matrix elements between singlet and triplet excited states were calculated by ORCA 5.0.4 using the B3LYP/G method with DKH2 DKH-def2-TZVP basis sets. Analysis was performed using Multiwfn 3.8[54,55].

### Hot-pressing and fiber fabrication methods

**Stamp.** TTPO (2.0 g) and guest (20 mg) were melted at 170 °C in a spoon. When the supercooled liquid reached 80 °C, an NUS mold stamp was pressed onto it. After removing the mold, an NUS afterglow glass stamp was obtained.

**Fiber.** TTPO (1.0 g) and guest (10 mg) were melted at 170 °C in a dish. The molten liquid was quickly transferred onto an 80 °C hot plate, from which a meter-scale fiber could be drawn from the supercooled liquid.

## Data availability

The data that support the findings of this study are available from the corresponding author B. Liu. The TTPO crystallographic data have been deposited at the Cambridge Crystallographic Data Centre (CCDC) under deposition numbers CCDC 2425812. These data can be obtained free of charge from the Cambridge Crystallographic Data Centre at https://www.ccdc.cam.ac.uk/structures/Search?Ccdcid=2425812&DatabaseToSearch=Published. Source data are provided with this paper.

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

## Acknowledgements

This study was supported by the Singapore National Research Foundation Investigatorship (A-8002259-00-00, B. Liu, NRFI09-0021, B. Liu), Tan Chin Tuan Centennial Professorship (E-467-00-0012-02, B. Liu).

## Author contributions

Y. Xue and B. Liu designed the project. Y. Xue performed all the experiments. Y. Xue and Z. Xie carried out the density-functional theory calculation, time-dependent density-functional theory calculation and spin orbit coupling matrix element calculation. Y. Xue, Z. Xie, Z. Yin, Y. Xu and B. Liu discussed the results and drafted the manuscript. B. Liu supervised the project. All authors contributed to the proofreading of the manuscript.

## Competing interests

The authors declare no competing interests.
