## [Transparent Peer Review file · Nature Communications]

Full-color processible afterglow organic small molecular glass

Corresponding Author: Professor Bin Liu

Version 0:

Reviewer comments:

Reviewer #1

(Remarks to the Author)

Molecular afterglow materials represent a promising research area with potential applications such as lighting, security, and electronics. In this work, Liu et al. reported a type of processible molecular glass derived from a small organic molecule. By doping with different guest molecules, they realized full-color (from violet to near-infrared) afterglow systems. They explained the underlying glass-forming process and afterglow mechanisms. Theoretical calculations are used to investigate the possible triplet energy transfer within these host-guest afterglow systems. They fabricated 3D objects and meter-long fibers using the developed glassy afterglow materials. This work is very interesting, and highlights the potential of glassy materials for advanced textiles and displays. To further improve the quality of the paper, I have outlined a few questions below:

1. Fig 1. lacks clarity and does not effectively illustrate the mechanism of molecular glassy through O-methylation from TPPO to TTPO. A more detailed representation of this process would improve its informativeness.
2. In Fig S8 and Fig S9, the fluorescence and delayed spectra are measured using different solvents. Could the authors clarify the rationale behind this choice? How might the solvent influence the observed spectra?
3. Compared to guest-doped polymer afterglow systems, what are the specific advantages of the guest-doped molecular glasses presented in this study? The authors could check and include related review recently (Sci. China Mater. 2024, 67, 3531).
4. It is surprising to observe the formation of the meter-long fibers from small organic molecules. Polymers can form fiber because of their long-chain structure, but what is the driving force that allows the small organic molecule to form such long fibers?
5. How stable is the glassy state of the small molecule TTPO? Since glassy formation is a kinetic process, will the glassy state transform into a crystalline state after some time?
6. Could the authors explain more about the photoactivation process before the afterglow emission?
7. Molecule-based glasses and their wide afterglow applications are hot topics among chemistry, materials and physics. To arouse a broad interest from readership in this field, some strongly related works on recent fabrications of molecular glasses systems (Nat. Commun. 2024, 15, 5519; Nat. Commun. 2023, 14, 1654; Angew. Chem. Int. Ed. 2023, 62, e202302751) and wide color-tunable afterglow applications (Nat. Commun. 2024, 15, 9491; Angew. Chem. Int. Ed. 2022, 61, e202208735; Mater. Horiz. 2024, 11, 4951) could be included as references.
8. The mechanism of triplet-triplet energy transfer is not fully elucidated in this research. Could authors provide more experimental evidence or data to support this proposed energy transfer mechanism?

Reviewer #2

(Remarks to the Author)

In this article, the authors report the development of a novel host molecule, tri(2-methylphenyl) phosphine oxide (TTPO), which retains the robust afterglow performance of TPPO while significantly reducing the crystallization rate. TTPO also exhibits temperature-dependent viscous supercooled liquid behavior. The authors' investigation of the luminescence mechanism reveals that TTPO not only provides a highly rigid microenvironment but also serves as an energy transfer bridge, enhancing afterglow generation. These attributes position TTPO as a promising platform for the rapid screening of phosphorescent molecules. The study offers valuable insights for researchers in related fields. However, the manuscript has several significant issues that need to be addressed and improved. Specific comments are as follows:

1. Purity significantly affects the afterglow luminescence properties. It is essential to verify the purity of all compounds,

particularly for those in the glassy state, using HPLC. In addition, a discussion of potential impurities and their impact on afterglow performance is necessary.

2. Differences in experimental parameters (e.g., slit width, pulse frequency) can influence the TRES spectra (Fig. 4b). The authors should ensure that all testing parameters remain consistent. Furthermore, it is recommended to prepare PMMA glassy samples using a uniform method, such as direct film preparation, and then compare their afterglow properties to those of evaporated films.

3. The rationale for selecting a doping ratio of 1 wt% is unclear. It is suggested to select a representative guest molecule and investigate the spectra and lifetimes at various doping ratios. Additionally, the manuscript should discuss trade-offs, such as the potential impact of higher dopant loading on material uniformity.

4. Theoretical computational data alone are insufficient to fully support the proposed mechanism in Fig. 4d. The authors should employ transient absorption spectroscopy to directly demonstrate the energy transfer process in the doped system.

5. Mechanical stress tests on fabricated fibers and films should be conducted to evaluate their strength and flexibility over extended periods. It is also crucial to assess whether these phosphorescent glasses can maintain consistent performance over time. The manuscript should include stability data to support long-term usability.

6. The manuscript lacks an in-depth exploration of environmental stability, such as the effects of oxygen and moisture quenching. These factors, along with long-term performance under real-world conditions, must be addressed to enhance the practical applicability of the materials.

7. The study would greatly benefit from the inclusion of prototypes or conceptual designs for real-world applications of TTPO, such as a proof-of-concept wearable sensor. This would provide a clearer demonstration of the material's potential utility.

Reviewer #3

(Remarks to the Author)

"Full-color processible afterglow organic small molecular glass" by Y. Xue et al. describes the synthesis of a novel molecular-glass forming material, TTPO, which can be used as a host material to activate room-temperature phosphorescence when doped with a variety of small molecule guests. The work includes a number of simulations to support its experimental data, and shows some very interesting processing of bulk samples, including the formation of doped molecular glass fibers. As I discuss in the comments below, there are a number of areas where the data and analysis should be strengthened and clarified, but I feel the main challenge for this work is the presence of a recent paper (<https://doi.org/10.1002/adma.202402478>) which demonstrates RTP in molecular aggregates embedded in polymer matrices. In general, the RTP performance and processability of these embedded aggregate-polymer systems are as good or better than those described here, and the work is not cited or discussed by the authors (which I certainly don't think is deliberate, as the paper mentioned above was published earlier this year). If the focus of this work is shifted more toward mechanistic insights/understanding instead of performance/novelty, I think it could be publishable (after the comments below are addressed), but I feel additional experimental support is needed to help establish some of the more mechanistic claims (i.e. that the TTPO triplet level can assist in intersystem crossing).

Specific Comments

1. In the first sentence of the introduction, it isn't clear what high signal-to-noise ratio refers to. This is perhaps in reference to bio-imaging applications where phosphorescence can out-last tissue auto-fluorescence and produce an image with high a signal-to-noise ratio? Additionally, afterglow materials are generally not common, and so I wouldn't agree that they possess "easy color tunability".

2. Recent work (<https://doi.org/10.1002/adma.202402478>) has shown aggregated small-molecule afterglow materials embedded in polymer matrices. These embedded aggregate systems address many of the limitations around processability and form factors of small molecules that the authors refer to in motivating their work. The polymer embedded aggregates also exhibit a broad range of emission colors and generally have longer lifetimes than those shown by Y. Xue et al. While the presence of this result does not necessarily undercut the small molecular glass approach employed by the authors, this result does need to be discussed so the current work can be put into the appropriate context.

3. Page 6, line 129 references "crystal analysis", and what appear to be unit cells are shown in Fig. 3a. It's unclear if these structures are purely simulated or if they come from x-ray measurements. This should be clarified.

4. The triplet energy of the new TTPO host is mentioned several times in the first several pages of the manuscript in the context of being large, but the value of the triplet is not directly mentioned until the discussion on page 10. I think including this value earlier on (or waiting to mention the triplet energy of the host) would be valuable

5. Because the presence of the violet afterglow from up-conversion in the PCA system will depend strongly on the triplet density (TTA scales as the triplet population squared), the relative intensity between the delayed fluorescence and the phosphorescence should be time variant. If the spectra in Fig. 3e are steady-state, that should be made clear, and if they are afterglow spectra that should also be indicated.

6. A number of the fluorescence spectra in Fig. S8 are quite broad and featureless, which is not typical of monomolecular fluorescence since there should be some vibronic character to the emission. Were concentration-dependent experiments done to confirm that this emission is not due to excimers?

7. The comparison of afterglows between the TTPO doped system and PMMA is another place where the results need to be put in the context of: <https://doi.org/10.1002/adma.202402478> In that work, the authors found that annealing to produce small aggregates massively increased RTP and lifetimes, and thus un-annealed small molecule – PMMA blends are not particularly representative of what performance can be achieved in that system.

8. The predicted ISC enhancement of having TTPO's triplet energy level sit between the S1 and T1 of most of the small molecule organics would benefit significantly from having some level of experimental validation. Otherwise, the presence or validity of this mechanism is unclear.

9. Page 11 lines 224 through 228 are totally separate from the rest of the paragraph and include no citations to back up the claims. It seems there is an argument being made that TTPO is synthesizable from cheap industrial waste products, but if this is true, the point should be made explicitly with references indicating where these waste products come from and how they would be turned into reagents for TTPO. As it is, there are two sentences discussing the beginning of this topic, and then the paragraph pivots directly into discussing hot pressing.

Minor Comments

1. Since two authors have the same initials Y. X., it may be appropriate to use full last names in the author contributions section to remove ambiguity.

2. Supplemental Figure S15—the x-axis should be “excitation power” instead of “exciton power”, and the label on inside the graph should say “slope”

3. Page 7, line 158: it is unclear what is meant by “both doping system”. Is “both” a typo perhaps?

4. The figure captions and figures in the manuscript and SI are often separated from one another, which makes the information more difficult to read. Since the manuscript would be reformatted prior to publication, those aren't as much of an issue, but it would be helpful to reformat the S.I. prior to any publication so that the figures and their captions are together on the same page.

5. There is some ambiguity around the use of “full-color” and “full-spectrum”. What is meant by this is that the host can be used with dopants that emit different colors, but at times it sounds like the emission is broadband or that an individual system can exhibit many colors. I think the piece would benefit from more concise language around this point.

Version 1:

Reviewer comments:

Reviewer #1

(Remarks to the Author)

In my view, the authors have answered all the questions and revised related points, and thus this revised work can be published as it is.

Reviewer #2

(Remarks to the Author)

I have carefully read the authors' responses to the reviewers' comments. I find the majority of the revisions satisfactory. However, a minor question remains. In Figures R8-R17, the authors present HPLC measurements. I am curious as to the reason for the significant variations in retention times observed across certain samples.

Reviewer #3

(Remarks to the Author)

Response to the rebuttal from the authors - reviewer #3

Overall, I am very impressed with the author's thorough response to the reviewer feedback from the first round. I feel they have gone above and beyond with the addition of several new experimental data points, which includes a 1:1 comparison with annealed PMMA films embedded with small molecules. In my view, the piece is publishable after responding to the comments below, but I do not need to see it again.

Specific comments:

1) Reviewer 1's comment #7 requests that the authors add 6 references to their work. I hope the editorial team will let the authors decide independently whether or not it is appropriate to cite this work without fear of a retaliatory rejection from reviewer 1.

2) In response to my 2nd comment, the authors brought up that the difference between evaporated and annealed PMMA films indicates that there is considerable batch-to-batch variation in this approach. This is also raised as one of several

advantages included in the new Table R1. I have several comments regarding this. Firstly, it is ambiguous what 'evaporated' is referring to in this context—presumably just the solvent being evaporated (typically called as-cast, or similar). But this should be stated more clearly, since <https://doi.org/10.1002/adma.202402478> also includes thermally evaporated neat films, which would seem to be a completely different fabrication method. Secondly, the fact that unannealed and annealed films perform differently is not indicative of batch-to-batch variation or process sensitivity as those are different processing conditions. To establish the advantage claimed by the authors, it must be shown that batches prepared using the same method produce different results (and that doped TTPO films prepared the same way perform consistently).

Reviewer #1(Remarks to the Author):

Molecular afterglow materials represent a promising research area with potential applications such as lighting, security, and electronics. In this work, Liu et al. reported a type of processable molecular glass derived from a small organic molecule. By doping with different guest molecules, they realized full-color (from violet to near-infrared) afterglow systems. They explained the underlying glass-forming process and afterglow mechanisms. Theoretical calculations are used to investigate the possible triplet energy transfer within these host-guest afterglow systems. They fabricated 3D objects and meter-long fibers using the developed glassy afterglow materials. This work is very interesting, and highlights the potential of glassy materials for advanced textiles and displays. To further improve the quality of the paper, I have outlined a few questions below:

1. Fig 1. lacks clarity and does not effectively illustrate the mechanism of molecular glassy through O-methylation from TPPO to TTPO. A more detailed representation of this process would improve its informativeness.

Response: According to the reviewer's suggestion. We have added more information about the o-methylation mechanism to Fig. R1.:

Fig. R1 (Fig. 1 in the revised manuscript) | Glass forming cycle and glass design strategy. a) Illustration of glass forming and aging. (i), Melting (ii), vitrification (iii), molecule reorganization. b) O-methylation design principle from TPPO to TTPO.

2. In Fig S8 and Fig S9, the fluorescence and delayed spectra are measured using different solvents. Could the authors clarify the rationale behind this choice? How might the solvent influence the observed spectra?

Response: In previous work, we used dichloromethane (DCM) as the solvent for fluorescence measurements. Meanwhile, toluene was chosen as the solvent for phosphorescence measurements at 77 K, owing to its non-polar nature and capability of forming rigid glassy solid, which helps to exhibit more phosphorescence fine structures.

For consistency, in the revised manuscript, we have used toluene as the solvent for all

the measurements.

Fig. R2 (Supplementary Fig. 14 in the revised SI.) | Fluorescence spectra of guests in toluene at room temperature (310 nm excitation, $[c] = 10^{-5} \text{ M}$)

3. Compared to guest-doped polymer afterglow systems, what are the specific advantages of the guest-doped molecular glasses presented in this study? The authors could check and include related review recently (*Sci. China Mater.* 2024, 67, 3531).

Response: We studied the referenced work (*Sci. China Mater.*, 2024, 67, 3531). Following the recommendations of Reviewer #3 (*Adv. Mater.*, 2024, 36, 2402478), we fabricated PMMA doping systems as representative polymer afterglow systems. To provide a comprehensive comparison, we systematically evaluated the TTPO doping systems and polymer doping systems from multiple perspectives. The summarized results are presented in the table below.

Table R1. Comparison between TTPO doping system and PMMA doping system

	TTPO doping system	PMMA doping system
Preparation methods	Melting-rapid cooling (156 °C)	Dissolving, mixing, evaporation, annealing (140 °C)
Time for preparation	seconds	hours
Free volume	Uniform	Diverse
Dopant dispersion	Uniform	Nonuniform due to dispersed free volume and irregular polymer tertiary structure
Batch difference	No batch-to-batch difference	Highly influenced by the fabrication methods

4. It is surprising to observe the formation of the meter-long fibers from small organic molecules. Polymers can form fiber because of their long-chain structure, but what is the driving force that allows the small organic molecule to form such long fibers?

Response: Due to the suitable viscosity of TTPO's supercooled liquid at 80 °C for thermal drawing and its glass transition temperature (T_g) at 25 °C, the material rapidly solidifies upon being drawn from the viscous supercooled state. The rapid solidification prevents further deformation that may be induced by viscous liquid flow. This is one of the key factors enabling the formation of long fibers.

On the other hand, an analysis of the crystal structure suggests the presence of multiple intermolecular interactions in TTPO, including C–H \cdots O (2.6-2.9 Å), C-H \cdots π (2.9 Å) and C-H \cdots H (2.4 Å) interactions. These relatively strong intermolecular interactions may also contribute to its fiber-forming ability.

5. How stable is the glassy state of the small molecule TTPO? Since glassy formation is a kinetic process, will the glassy state transform into a crystalline state after some time?

Response: The glassy state of the small molecule TTPO exhibits some stability under ambient conditions (*Mol. Pharmaceutics*, **2021**, *18*, 278; *J. Pharm. Sci.*, **2010**, *99*, 3787.). To investigate this, we collected p-XRD data and corresponding images of TTPO samples, which indicate that TTPO gradually undergoes crystallization around 18 hours (Fig. R3).

To improve the glass stability of the host material, strategies such as enhancing intermolecular interactions through hydrogen bonding or ionic bonding, increasing molecular weight, and further restricting conformational flexibility could be employed, which is currently ongoing in our laboratory.

**Fig. R3 (Supplementary Fig. 5 in the revised SI.)** | Time-dependent p-XRD

measurements of TTPO films.

6. Could the authors explain more about the photoactivation process before the afterglow emission?

Response: The discussion about the photoactivation process “We found a photoactivation behavior when we test the RTP performance of glassy films. Phosphorescence remains inactive without photoactivation due to the inevitable presence of residual oxygen in the system during the fabrication of glass films in the air. As a result, upon continuous irradiation for 10 seconds, the triplet emission component in the TNpP@TTPO doping system significantly increases. This process was captured through time-dependent PL measurements (Fig. R4 a,b). After photoactivation, clear and bright afterglow could be observed from the TNpP@TTPO system in its amorphous form, as confirmed by p-XRD experiments (Fig. 3c). This phenomenon can be attributed to the generated triplet excitons from the guest molecules gradually consume the residual ground-state oxygen, thereby activating the phosphorescence (Fig. R4c).” has been incorporated into the main text on Page7 line 143 to 151. This discussion could provide a clear explanation of the oxygen-quenching effect and the mechanism of photoactivation.

Fig. R4 (Supplementary Fig. 7 in the revised SI.) | a) Time-dependent PL spectra of the TNpP@TTPO doping system (excited at 310 nm to minimize interference from the excitation source). b) Time-dependent emission intensity at 528 nm. c) Schematic illustration of the photoactivation process.

7. Molecule-based glasses and their wide afterglow applications are hot topics among chemistry, materials and physics. To arouse a broad interest from readership in this field, some strongly related works on recent fabrications of molecular glasses systems (Nat. Commun. 2024, 15, 5519; Nat. Commun. 2023, 14, 1654; Angew. Chem. Int. Ed. 2023, 62, e202302751) and wide color-tunable afterglow applications (Nat. Commun. 2024, 15, 9491; Angew. Chem. Int. Ed. 2022, 61, e202208735; Mater. Horiz. 2024, 11, 4951) could be included as references.

Response: We sincerely appreciate the reviewer’s insightful comments. We have included the references in the introduction on Page 3, Line 59.

8. The mechanism of triplet-triplet energy transfer is not fully elucidated in this research. Could authors provide more experimental evidence or data to support this proposed energy transfer mechanism?

Response: Following the reviewer's suggestion, we carefully evaluated the luminescence properties of the TTPO film. From the absorption spectrum, the TTPO film exhibited a weak absorption at 365 nm, indicating that it is excitable under the excitation conditions of our doping system (Fig. R5a).

Photophysical studies revealed that the TTPO film exhibits fluorescence at 400 nm and weak but detectable phosphorescence at room temperature after photoactivation, with a maximum emission at 530 nm. To further investigate the triplet-triplet energy transfer (TTET) process, we measured the lifetime at 450 nm of the TTPO film (42 ms). Then compared this with the emission lifetimes at 450 nm across various doping systems, where the guest phosphorescence spectra do not exhibit emission at this wavelength (Fig. R5b). Taking BrPhBd@TTPO as an example, the lifetime of BrPhBd@TTPO at 450 nm is 6 ms, which is significantly shorter compared to TTPO itself. This shortened lifetime serves as strong evidence supporting the TTET process (Fig. R5c).

Fig. R5 (Supplementary Fig. 24 in the revised SI. and Fig. 4c in the revised manuscript) | a) Absorption and PL of TTPO amorphous film. b) The overlap of delay 8ms spectra of TTPO film and BrPhBd@TTPO doping system. c) The lifetime at 450 nm of TTPO film and BrPhBd@TTPO doping system.

Similarly, other doping systems, including 1,8NA@TTPO, DPNPA@TTPO, Py@TTPO, and DiBrBDP@TTPO, exhibit the same trend of reduced lifetime at 450 nm compared to the TTPO film, further confirming the occurrence of TTET (Fig. R6). Therefore, we modified the luminescence mechanism to Fig. R7.

Fig. R6 (Supplementary Fig. 26 in the revised SI.) | a,c,e,g) Overlap of the delay 8 ms spectra of TTPO films and doping systems. b,d,f,h) The lifetime at 450 nm of TTPO film and dopants@TTPO doping system.

Fig. R7 (Fig. 4d in the revised manuscript) | Jablonski diagram for proposed photophysical processes of doping systems.

Reviewer #2 (Remarks to the Author):

In this article, the authors report the development of a novel host molecule, tri(2-methylphenyl) phosphine oxide (TTPO), which retains the robust afterglow performance of TPPO while significantly reducing the crystallization rate. TTPO also exhibits temperature-dependent viscous supercooled liquid behavior. The authors' investigation of the luminescence mechanism reveals that TTPO not only provides a highly rigid microenvironment but also serves as an energy transfer bridge, enhancing afterglow generation. These attributes position TTPO as a promising platform for the rapid screening of phosphorescent molecules. The study offers valuable insights for researchers in related fields. However, the manuscript has several significant issues that need to be addressed and improved. Specific comments are as follows: 1. Purity significantly affects the afterglow luminescence properties. It is essential to verify the purity of all compounds, particularly for those in the glassy state, using HPLC. In addition, a discussion of potential impurities and their impact on afterglow performance is necessary.

Response: Following the reviewer's suggestion, we verified the purity of all compounds, including those in the glassy state. HPLC measurements confirm that all compounds are pure and remain stable without degradation during glass formation (Fig. R8-17). However, due to the weak absorption of TNpP, we were unable to detect the TNpP signal from the 1 wt% TNpP@TTPO doping system, even when the stock solution concentration for HPLC injection reached 100 ppm.

Fig. R8 | HPLC measurements of MAMOBz and MAMOBz@TTPO doping system (in glassy state)

Fig. R9 | HPLC measurements of CM1 and CM1@TTPO doping system (in glassy state)

Fig. R10 | HPLC measurements of 7HBCz and 7HBCz@TTPO doping system (in glassy state)

Fig. R11 | HPLC measurements of 1,8NA and 1,8NA@TTPO doping system (in glassy state)

Fig. R12 | HPLC measurements of 5HBCz and 5HBCz@TTPO doping system (in glassy state)

Fig. R13 | HPLC measurements of DPNPA and DPNPA@TTPO doping system (in glassy state)

Fig. R14 | HPLC measurements of BrPhBd and BrPhBd@TTPO doping system (in glassy state)

Fig. R15 | HPLC measurements of Py and Py@TTPO doping system (in glassy state)

Fig. R16 | HPLC measurements of PCA and PCA@TTPO doping system (in glassy state)

Fig. R17 | HPLC measurements of DiBrBDP and DiBrBDP@TTPO doping system (in glassy state)

2. Differences in experimental parameters (e.g., slit width, pulse frequency) can influence the TRES spectra (Fig. 4b). The authors should ensure that all testing parameters remain consistent. Furthermore, it is recommended to prepare PMMA glassy samples using a uniform method, such as direct film preparation, and then compare their afterglow properties to those of evaporated films.

Response: We sincerely appreciate the reviewer's insightful comments. We acknowledge that differences in experimental parameters, such as slit width and pulse frequency, can influence the time-resolved emission spectra (TRES). To ensure consistency, we carefully controlled these parameters during our measurements, maintaining identical settings across all experiments.

CCD model and integral time: Ocean Optic QE 65 Pro spectrometer, integral time 8 ms.

Optical fiber: 600 μ m diameter with reflection probe.

Excitation source: 365 nm LED light source, output power 10 mW (600 μ m diameter optic fiber), FWHM < 13 nm

In response to Reviewer #3's suggestion, we re-evaluated the PMMA doping system using annealing methods (<https://doi.org/10.1002/adma.202402478>). Our results show that PMMA doping systems fabricated via the annealing method exhibit significantly improved performance compared to those prepared by evaporated films. However, their afterglow performance remains weaker than that of TTPO-based systems.

To further investigate this, we conducted TRES experiments and used a camera to capture their afterglow performance. The TTPO and PMMA doping systems were placed under identical excitation conditions, with both samples irradiated simultaneously using a large-area 365 nm UV lamp (100 W) to ensure uniform excitation. The comparative analysis, along with the TRES data, is provided in Fig. R27.

Fig. R27 (Supplementary Fig. 23 in revised SI) | TRES and afterglow images of dopants@TTPO vs dopants@PMMA.

3. The rationale for selecting a doping ratio of 1 wt% is unclear. It is suggested to select a representative guest molecule and investigate the spectra and lifetimes at various doping ratios. Additionally, the manuscript should discuss trade-offs, such as the potential impact of higher dopant loading on material uniformity.

Response: Among all TNpP doping systems with varying doping ratios, the TRES and lifetime measurements at 527 nm indicate that TNpP@TTPO at 1 wt% exhibited the best afterglow performance (Fig.R18, R19). Therefore, all doping systems in this study were prepared using 1 wt% doping ratio for consistency.

The low viscosity melt and small molecular size of TTPO enable the efficient dissolution

and uniform dispersion of most organic molecules. During sample preparation, stirring the molten TTPO with the dopants was essential to ensure thorough dissolution and well dispersion. Even with a dopant loading of 10 wt%, rapidly cooling the mixture to room temperature still resulted in the formation of a transparent and uniform glassy film. However, at higher concentrations (> 10 wt%), the phosphorescence signal was reduced due to concentration quenching.

Fig. R18 (Supplementary Fig. 11 in the revised SI) | Afterglow photographs and time-resolved phosphorescence spectra of TNp@TTPO with varying doping ratios.

Fig. R19 (Supplementary Fig. 12 in the revised SI) | Delay 8 ms and decay profile at 527 nm of TNp@TTPO with varying doping ratios.

4. Theoretical computational data alone are insufficient to fully support the proposed mechanism in Fig. 4d. The authors should employ transient absorption spectroscopy to directly demonstrate the energy transfer process in the doped system.

Response: We sincerely appreciate the reviewer's valuable question. Due to limitations in the experimental conditions, we were unable to conduct the transient absorption by using fs or ns transient absorption spectroscopy. In alternative ways, we carefully studied the photophysical properties of TTPO films and validated the TTET process via lifetime comparison and microsecond TRES measurement.

We carefully evaluated the luminescence properties of the TTPO film. From the

absorption spectrum, the TTPO film exhibited a weak absorption at 365 nm, indicating that it is excitable under the excitation conditions of our doping system (Fig. R5a). Photophysical studies revealed that the TTPO film exhibits fluorescence at 400 nm and weak but detectable phosphorescence at room temperature after photoactivation, with a maximum emission at 530 nm. To investigate the triplet-triplet energy transfer process, we measured the lifetime at 450 nm of the TTPO film (42 ms). Then compared this with the emission lifetimes at 450 nm across various doping systems, where the guest phosphorescence spectra do not exhibit emission at this wavelength (Fig. R5b). Taking BrPhBd@TTPO as an example, the lifetime of BrPhBd@TTPO at 450 nm is 6 ms, which is significantly shorter compared to TTPO itself. This shortened lifetime serves as strong evidence supporting the TTET process (Fig. R5c).

Fig. R5 (Supplementary Fig. 24 in the revised SI. and Fig. 4c in the revised manuscript)
 | a) Absorption and PLof TTPO amorphous film.b) The overlap of delay 8ms spectra of TTPO film and BrPhBd@TTPO doping system. c) The lifetime at 450 nm of TTPO film and BrPhBd@TTPO doping system.

Similarly, other doping systems, including 1,8NA@TTPO, DPNPA@TTPO, Py@TTPO, and DiBrBDP@TTPO, exhibit the same trend of reduced lifetime at 450 nm compared to the TTPO fim, further confirming the occurrence of TTET (Fig. R6).

Fig. R6 (Supplementary Fig. 26 in the revised SI.) | a,c,e,g) Overlap of the delay 8 ms spectra of TTPO films and doping systems. b,d,f,h) The lifetime at 450 nm of TTPO film and dopants@TTPO doping system.

Furthermore, we conducted microsecond TRES experiment for BrPhBd@TTPO doping system. From Fig. R20, the emission peak at 0.2 ms is attributed to the triplet emission of TTPO as its emission profile is identical to that of TTPO film. Between 0.2 ms and 0.4 ms, the emission undergoes a gradual redshift and broadening. From 0.4 ms to 20 ms, the emission intensity from the TTPO triplet state progressively diminishes, while the triplet emission from BrPhBd gradually increases. This dynamic evolution provides strong evidence supporting the TTET process. Therefore, we modified the luminescence mechanism to Fig. R7.

Fig. R20 (Supplementary Fig.25 in the revised SI) | a) Microsecond TRES experiment of BrPhBd@TTPO doping system. b) Sliced emission spectra within 50 ms.

Fig. R7 (Fig. 4d in the revised manuscript) | Jablonski diagram for proposed photophysical processes of doping systems.

5. Mechanical stress tests on fabricated fibers and films should be conducted to evaluate their strength and flexibility over extended periods. It is also crucial to assess whether these phosphorescent glasses can maintain consistent performance over time. The manuscript should include stability data to support long-term usability.

Response: The mechanical properties of the films in both amorphous and crystalline forms were evaluated using microindentation and the fibers were evaluated using tensile strength–displacement tests (Fig. R21).

For the glassy film, the indentation hardness (HIT) and indentation modulus (EIT) were measured to be 15.23 MPa (mean value) and 0.75 GPa (mean value), respectively. After crystallization, the film became softer, with a reduced hardness of 7.87 MPa (mean value), while the indentation modulus remained at 0.70 GPa (mean value).

The higher hardness in the amorphous state may be attributed to its disordered structure, which provides stronger intermolecular constraints. In the glassy state, molecules are arranged in a random, disordered manner, leading to a uniform distribution in different directions. This could help restrict structural slippage under external force and thereby enhancing hardness. In contrast, in the crystalline state, molecules adopt a more ordered arrangement, which introduces slip planes within the

layers or crystal lattice. These slip planes may reduce the directional strength of intermolecular interactions, leading to lower hardness and increased susceptibility to deformation under stress.

For the fibers, the elastic modulus was measured at 1.0 GPa for the amorphous fiber and 3.1 GPa for the crystalline fiber, indicating that the amorphous fiber is significantly more flexible than its crystalline counterpart. Regarding mechanical strength and failure behavior, the crystalline fiber fractures at an early stage, failing at a strain of ~ 0.005 and a stress of 16 MPa, which suggests brittle failure, meaning the material cannot undergo significant deformation before breaking. In contrast, the amorphous fiber continues to deform up to ~ 0.02 strain before failing at a stress of 21 MPa, demonstrating higher ductility.

The difference between amorphous and crystalline fibers can be explained by the molecular arrangement: the disordered structure of the amorphous fiber enables more uniform stress distribution, which contributes to its increased flexibility and greater strain accommodation, allowing it to sustain more deformation before failure.

Fig. R21 (Supplementary Fig.46 in the revised SI) | Mechanical properties of films and fibers. a,b) Mircoindentation measurement of glassy film and crystalline film. c) Tensile strength–displacement curve of amorphous fiber and crystalline fiber.

To evaluate the long-term phosphorescence properties, we took the TNpP@TTPO doping system as an example to explore. The results show that the phosphorescence intensity remained unchanged over 9 weeks (Fig. R22a,b). Additionally, we also evaluated the phosphorescence properties under different morphological states and compared the delayed spectra and lifetime at 528 nm for both crystalline and glassy films. The phosphorescence properties remained unchanged across different morphological states (Fig. R22c,d).

Fig. R22 (Supplementary Fig.10 in the revised SI) | a) Phosphorescence spectra at different weeks. **b)** Phosphorescence intensity over 9 weeks. **c)** The decay 8 ms spectra and photo of TNpP@TTPO glassy and crystalline films. **d)** Phosphorescence lifetime of TNpP@TTPO glassy and crystalline films at 528 nm (Camera setting: ISO 400 for daylight imaging, ISO 5000 for phosphorescence imaging).

6. The manuscript lacks an in-depth exploration of environmental stability, such as the effects of oxygen and moisture quenching. These factors, along with long-term performance under real-world conditions, must be addressed to enhance the practical applicability of the materials.

Response: We sincerely appreciate the reviewer's valuable suggestions. To investigate the environmental stability of TTPO doping systems, the TNpP@TTPO films were immersed in pure water and exposed to a pure oxygen atmosphere respectively. After photoactivation, the TNpP@TTPO films exhibited bright phosphorescence with a duration of approximately 1.5 seconds in the water or oxygen. Notably, even after being submerged in water for three days, the afterglow performance remained largely unaffected (Fig. R23). This further demonstrates that the TTPO film can effectively block against quenching factors such as oxygen and humidity.

Fig.R23 (Supplementary Fig. 9 in the revised SI) | Afterglow properties of TNpP@TTPO in water and oxygen.

7. The study would greatly benefit from the inclusion of prototypes or conceptual designs for real-world applications of TTPO, such as a proof-of-concept wearable sensor. This would provide a clearer demonstration of the material's potential utility.

Response: We sincerely appreciate the reviewer's valuable suggestions. To fully leverage the excellent processability of the TTPO doping system and unlock its potential applications, we explored its feasibility for optical waveguide fiber fabrication. A 7 cm long fiber was thermal drawn from the viscous supercool liquid of TNpP@TTPO system. Upon UV excitation at either the tip or the middle of the fiber, the fiber emits light at the incident end, with the emission propagating along the fiber axis, demonstrating waveguiding behavior. After the excitation source is removed, the fiber continues to emit light at its ends, which can be easily observed with the naked eye (Fig. R24). Bright afterglow optical fibers are rarely reported, and in this system, TTPO serves as a transparent fiber host while the guest molecules provide luminescence properties. By modifying the phosphorescent dopants, the afterglow fiber characteristics can be further tuned and customized, offering a more flexible approach to exploring the potential applications of afterglow optical fibers in areas

such as optical signal transmission, flexible photonic devices, and nighttime displays.

Fig. R24 (Supplementary Fig. 47 in the revised SI) | Demonstration of afterglow optical waveguide fiber.

Reviewer #3 (Remarks to the Author)

“Full-color processible afterglow organic small molecular glass” by Y. Xue et al. describes the synthesis of a novel molecular-glass forming material, TTPO, which can be used as a host material to activate room-temperature phosphorescence when doped with a variety of small molecule guests. The work includes a number of simulations to support its experimental data, and shows some very interesting processing of bulk samples, including the formation of doped molecular glass fibers. As I discuss in the comments below, there are a number of areas where the data and analysis should be strengthened and clarified, but I feel the main challenge for this work is the presence of a recent paper (<https://doi.org/10.1002/adma.202402478>) which demonstrates RTP in molecular aggregates embedded in polymer matrices. In general, the RTP performance and processability of these embedded aggregate-polymer systems are as-good or better than those described here, and the work is not cited or discussed by the authors (which I certainly don't think is deliberate, as the paper mentioned above was published earlier this year). If the focus of this work is shifted more toward mechanistic insights/understanding instead of performance/novelty, I think it could be publishable (after the comments below are addressed), but I feel additional experimental support is needed to help establish some of the more mechanistic claims (i.e. that the TTPO triplet level can assist in intersystem crossing).

Specific Comments

1. In the first sentence of the introduction, it isn't clear what high signal-to-noise ratio refers to. This is perhaps in reference to bio-imaging applications where phosphorescence can out-last tissue auto-fluorescence and produce an image with high a signal-to-noise ratio? Additionally, afterglow materials are generally not common, and so I wouldn't agree that they possess “easy color tunability”.

Response: We sincerely appreciate the reviewer's valuable suggestions. The sentence has been modified to *“Organic afterglow materials, capable of maintaining luminescence for over several tens of milliseconds or even seconds after the excitation ceases, have garnered significant attention in recent years for potential applications in fields such as multicolor display, encryption, bioimaging, and responsive sensors.”* on Page 2, line 33 to 35.

2. Recent work (<https://doi.org/10.1002/adma.202402478>) has shown aggregated small-molecule afterglow materials embedded in polymer matrices. These embedded aggregate systems address many of the limitations around processability and form factors of small molecules that the authors refer to in motivating their work. The polymer embedded aggregates also exhibit a broad range of emission colors and generally have longer lifetimes than those shown by Y. Xue et al. While the presence of this result does not necessarily undercut the small molecular glass approach

employed by the authors, this result does need to be discussed so the current work can be put into the appropriate context.

Response: We sincerely appreciate the reviewer's valuable suggestions. We recognize the significance of polymer-embedded aggregate systems and agree that they almost don't have processability challenges. However, our small molecular glass approach offers distinct advantages, such as well-defined structure with no batch-to-batch variation, rapid sample preparation in minutes, and uniform dopant dispersion, etc, which are very difficult to achieve with polymer systems. For example, by comparing the evaporated PMMA doping system and the annealed PMMA doping system (<https://doi.org/10.1002/adma.202402478>), one can conclude that fabrication methods significantly impact the afterglow performance of polymeric doping systems, leading to batch-to-batch variations.

Therefore, these two strategies are complementary rather than directly competing, and our study contributes to expanding the toolbox for designing high-performance and processible afterglow materials in the small molecular system.

3. Page 6, line 129 references "crystal analysis", and what appear to be unit cells are shown in Fig. 3a. It's unclear if these structures are purely simulated or if they come from x-ray measurements. This should be clarified.

Response: We acknowledge that this information was missing in the main text. The crystal structure is from x-ray measurements.

Based on our work, TTPO was crystallized from triethylamine, forming a monoclinic crystal system with space group $P2_1/c$. The asymmetric unit comprises a TTPO with the molecular formula of $C_{21}H_{21}OP$. One of the *o*-tolyl groups is disordered into two positions with an occupancy ratio of 78:22 which corresponds to the *exo3* and *exo2* conformations (Fig. R25). The crystal structure has been deposited on the CCDC website under the deposition number 2425812.

Fig. R25 (Supplementary Fig. 6 in the revised SI) | Crystal information of TTPO.

Table R2 (Table S2 in the revised SI) Crystal data and structure refinement for single crystal of TTPO

Formula	C ₂₁ H ₂₁ OP	$\rho_{\text{calc}}/\text{g cm}^{-3}$	1.226
Formula weight	320.35 g/mol	Z	4
Temperature / K	100 K	Absorption coefficient	1.402
Crystal system	Monoclinic	F (000)	680.0
Space group	P2 ₁ /c	Crystal size/mm³	0.161 × 0.114 × 0.076
a / Å	15.9602(5)	Wavelength / Å	1.54178
b / Å	7.7521(2)	Reflections collected	0.0422 (2734)
c / Å	14.2751(4)	Data/ restraints/ parameters	3065/ 277/ 276
α / °	90	Independent reflections	3065
			[R _{int} = 0.0497, R _{sigma} = 0.0296]
β / °	100.661(2)	Goodness-of-fit on F2	1.100
γ / °	90	R₁, ^[a] wR₂ ^[b] [$I \geq 2\sigma(I)$]	R ₁ = 0.0422, wR ₂ = 0.0942
Volume / Å³	1735.70(9)	R₁, wR₂ [all data]	R ₁ = 0.0485, wR ₂ = 0.0997

4. The triplet energy of the new TTPO host is mentioned several times in the first several pages of the manuscript in the context of being large, but the value of the triplet is not directly mentioned until the discussion on page 10. I think including this value earlier on (or waiting to mention the triplet energy of the host) would be valuable.

Response: We have added the triplet energy value 2.95 eV at the very beginning of the section (Page 5, line 97) where we first discuss the triplet energy of TTPO.

5. Because the presence of the violet afterglow from up-conversion in the PCA system will depend strongly on the triplet density (TTA scales as the triplet population squared), the relative intensity between the delayed fluorescence and the phosphorescence should be time variant. If the spectra in Fig. 3e are steady-state, that should be made clear, and if they are afterglow spectra that should also be indicated.

Response: We sincerely appreciate the reviewer's valuable suggestions. The relationship between up-conversion fluorescence intensity and excitation power was established based on the delayed 8 ms spectra.

The spectra shown in Fig. 3e represent the delayed 8 ms spectra of different doping systems. The missing information has now been added to the main text.

6. A number of the fluorescence spectra in Fig. S8 are quite broad and featureless, which is not typical of monomolecular fluorescence since there should be some vibronic character to the emission. Were concentration-dependent experiments done to confirm that this emission is not due to excimers?

Response: We sincerely appreciate the reviewer's valuable suggestions. We conducted concentration-dependent experiments on CM1, TNpP, MAMOBz, and DPNPA in dichloromethane, with concentrations ranging from 10⁻¹⁰ M to 10⁻⁵ M (Fig. R26). However, the fluorescence spectra of these compounds remained unchanged across all concentrations. Therefore, we conclude that their broad and featureless fluorescence is an intrinsic characteristic rather than a result of excimer formation.

Fig. R26 | Fluorescence spectra of CM1, TNpP, MAMOBz and DPNPA in various concentrations (DCM as solvent, 330 nm excitation)

7. The comparison of afterglows between the TTPO doped system and PMMA is another place where the results need to be put in the context of: <https://doi.org/10.1002/adma.202402478> In that work, the authors found that annealing to produce small aggregates massively increased RTP and lifetimes, and thus un-annealed small molecule – PMMA blends are not particularly representative of what performance can be achieved in that system.

Response: We sincerely appreciate the reviewer’s valuable suggestions. Following the reviewer’s suggestion, we purchased the same batch of PMMA from Sigma (<https://www.sigmaaldrich.com/SG/en/product/aldrich/445746>) as reported in the literature and attempted to prepare the doping system accordingly. To enable a direct comparison, we selected 9-DT@PMMA, the best-performing doping system, and successfully reproduced its excellent afterglow performance using the reported method (Fig. R27a).

Subsequently, 9-DT was doped into TTPO using the melt-cooling method. The TRES experiment confirmed that TTPO exhibited better afterglow performance compared to PMMA-based systems. Following this experiment, other dopants in this study were incorporated into PMMA using the same methods, with the corresponding TRES data collected. Additionally, both dopants@PMMA and dopants@TTPO were simultaneously excited using a large-area 100 W UV lamp, and their afterglow images were recorded using a camera (Fig. R27b-j). Through TRES and image comparison, we found that the TTPO doping system demonstrated better afterglow performance than the PMMA doping system.

Fig. R27 (Supplementary Fig. 23 in revised SI) | TRES and afterglow images of dopants@TTPO vs dopants@PMMA.

8. The predicted ISC enhancement of having TTPO's triplet energy level sit between the S1 and T1 of most of the small molecule organics would benefit significantly from having some level of experimental validation. Otherwise, the presence or validity of this mechanism is unclear.

Response: We sincerely appreciate the reviewer's valuable suggestions. The ISC enhancement conclusion was inspired by the other researcher's works (*Angew. Chem. Int. Ed.*, **2020**, *59*, 16054; *Angew. Chem. Int. Ed.*, **2023**, *62*, e202315911; *Angew. Chem. Int. Ed.*, **2024**, *63*, e202319089), where the methodologies involved comparing the singlet and triplet energy levels between the host and guest, followed by theoretical calculations. Following this approach, we conducted energy level comparisons and modelling experiments. The modelling results indicated the presence of additional ISC channels in the TTPO and guest pairs. This may promote the afterglow performance (**Supplementary Fig. 27–42**).

Through a comprehensive study of TTPO photophysical properties, lifetime comparisons and microsecond TRES experiments, we successfully validated the TTET process from TTPO to the guest (**Fig. R5,6,20**).

9. Page 11 lines 224 through 228 are totally separate from the rest of the paragraph and include no citations to back up the claims. It seems there is an argument being made that TTPO is synthesizable from cheap industrial waste products, but if this is true, the point should be made explicitly with references indicating where these waste products come from and how they would be turned into reagents for TTPO. As it is,

there are two sentences discussing the beginning of this topic, and then the paragraph pivots directly into discussing hot pressing.

Response: We sincerely appreciate the reviewer's valuable suggestions. The related discussion has been removed from the context.

Minor Comments

1. Since two authors have the same initials Y. X., it may be appropriate to use full last names in the author contributions section to remove ambiguity.

Response: We have used full last names for all authors in the author contributions section.

2. Supplemental Figure S15—the x-axis should be “excitation power” instead of “exciton power”, and the label on inside the graph should say “slope”

Response: We have changed the typos in Figure S15 (**Supplementary Fig. 21 in revised SI**).

3. Page 7, line 158: it is unclear what is meant by “both doping system”. Is “both” a typo perhaps?

Response: We have deleted “both”.

4. The figure captions and figures in the manuscript and SI are often separated from one another, which makes the information more difficult to read. Since the manuscript would be reformatted prior to publication, those aren't as much of an issue, but It would be helpful to reformat the S.I. prior to any publication so that the figures and their captions are together on the same page.

Response: We have reformatted the SI.

5. There is some ambiguity around the use of "full-color" and "full-spectrum". What is meant by this is that the host can be used with dopants that emit different colors, but at times it sounds like the emission is broadband or that an individual system can exhibit many colors. I think the piece would benefit from more concise language around this point.

Response: We have modified the sentence to “However, developing materials that exhibit long-lasting, full-color afterglow from violet to near-infrared while achieving high phosphorescence quantum yields remains challenging.” on page 2, line 38.

Thank you for your time, and we look forward to your final decision in due course.

Sincerely,

Prof. Bin Liu
Department of Chemical and Biomolecular Engineering
National University of Singapore
Singapore, 117585

Reviewer #1 (Remarks to the Author):

In my view, the authors have answered all the questions and revised related points, and thus this revised work can be published as it is.

Response: We sincerely appreciate the reviewer for the positive feedback.

Reviewer #2 (Remarks to the Author):

I have carefully read the authors' responses to the reviewers' comments. I find the majority of the revisions satisfactory. However, a minor question remains. In Figures R8-R17, the authors present HPLC measurements. I am curious as to the reason for the significant variations in retention times observed across certain samples.

Response: We acknowledge that the elution conditions were not clearly described in the main text. The specific elution conditions are provided in Figures R8-R17. As different solvent combinations are needed for different guest molecules, therefore, TTPO shows different retention time under different solvent combinations.

Fig. R1 | HPLC measurements of TTPO under different elution conditions. a) Elution condition for MAMOBz@TTPO doping system, CM1@TTPO doping system, 7HBCz@TTPO doping system, TNpP@TTPO doping system, 5HBCz@TTPO doping system, 1,8 NA@TTPO doping system. b) Elution condition for DPNPA@TTPO doping system, BrPhBd@TTPO doping system, Py@TTPO doping system. c) Elution condition for PCA@TTPO doping system. d) Elution condition for DiBrBDP@TTPO doping system.

Reviewer #3 (Remarks to the Author):

Response to the rebuttal from the authors - reviewer #3

Overall, I am very impressed with the author's thorough response to the reviewer feedback from the first round. I feel they have gone above and beyond with the addition of several new experimental data points, which includes a 1:1 comparison with annealed PMMA films embedded with small molecules. In my view, the piece is publishable after responding to the comments below, but I do not need to see it again.

Specific comments:

1) Reviewer 1's comment #7 requests that the authors add 6 references to their work. I hope the editorial team will let the authors decide independently whether or not it is appropriate to cite this work without fear of a retaliatory rejection from reviewer 1.

Response: We have carefully studied the references recommended by Reviewer 1. These references discuss the formulation and applications of amorphous room-temperature afterglow materials based on supramolecular glasses. As they are relevant to our work, we have selected the two most pertinent references for citation (*Sci. China Mater.*, **2024**, *67*, 3531–3536; *Nat. Commun.*, **2024**, *15*, 9491).

2) In response to my 2nd comment, the authors brought up that the difference between evaporated and annealed PMMA films indicates that there is considerable batch-to-batch variation in this approach. This is also raised as one of several advantages included in the new Table R1. I have several comments regarding this. Firstly, it is ambiguous what 'evaporated' is referring to in this context—presumably just the solvent being evaporated (typically called as-cast, or similar). But this should be stated more clearly, since <https://doi.org/10.1002/adma.202402478> also includes thermally evaporated neat films, which would seem to be a completely different fabrication method. Secondly, the fact that unannealed and annealed films perform differently is not indicative of batch-to-batch variation or process sensitivity as those are different processing conditions. To establish the advantage claimed by the authors, it must be shown that batches prepared using the same method produce different results (and that doped TTPO films prepared the same way perform consistently).

Response: We sincerely appreciate the reviewer's valuable suggestion. Firstly, the "evaporated film" refers to a film prepared by allowing the solvent to evaporate naturally without any post-annealing treatment. This fabrication method can lead to variations in the residual solvent content, which in turn results in differences in the afterglow performance of the PMMA doping films—hence contributing to the observed batch-to-batch variation.

The work (<https://doi.org/10.1002/adma.202402478>) established a standardized processing method and we were able to independently reproduce their results. Therefore, we believe the statement that the batch-to-batch difference in PMMA doping system is not appropriate. We have accordingly revised the table as shown below (**Table R1**).

However, we would like to emphasize that less batch-to-batch difference is one of the key advantages of the molecular glass over polymer. This statement is supported by several references (*Advanced Materials*. De Gruyter, Berlin, Boston, 2020.; *Adv. Mater.* **2008**, *20*, 3355–3361 etc.).

Table R1. Comparison between TTPO doping system and PMMA doping system

	TTPO doping system	PMMA doping system
Preparation methods	Melting-rapid cooling (156 °C)	Dissolving, mixing, evaporation, annealing (140 °C)
Time for preparation	Seconds	Hours
Free volume	Uniform	Diverse
Dopant dispersion	Uniform	May not be uniform due to dispersed free volume and irregular polymer tertiary structure

Thank you for your time, and we look forward to your final decision in due course.

Sincerely,

Prof. Bin Liu

Department of Chemical and Biomolecular Engineering

National University of Singapore

Singapore, 117585